# Comparison of visual performance between bifocal and extended-depth-of-focus intraocular lenses

Hitoshi Tabuchi[1,2�], Hirotaka Tanabe[1�]*, Tomoki Shirakami[1], Kosuke Takase[1], Tomohiro Shojo[1], Tomofusa Yamauchi[1]

**1** Department of Ophthalmology, Tsukazaki Hospital, Himeji, Japan, **2** Department of Technology and Design Thinking for Medicine, Hiroshima University Graduate School of Biomedical and Health Sciences, Hiroshima, Japan

☊ These authors contributed equally to this work.
* tennsyoudragon@icloud.com

**Data Availability Statement:** All relevant data are within the manuscript and its Supporting Information files.

## Abstract

We compared the visual performance of a bifocal intraocular lens (IOL) (ZMB00) and an extended-depth-of-focus (EDOF) IOL (ZXR00V) by evaluating postoperative parameters at 10 weeks after the last surgery in cataract patients who underwent bilateral ZMB00 or ZXR00V implantation between 2011 and 2020. The right and left lenses were implanted within 3 months of each other. The study enrolled 1536 eyes of 768 patients; the ZMB00 group comprised 1326 eyes of 663 patients (age: 67.0 ± 7.8 years; female/male, 518/145), and the ZXR00V group comprised 210 eyes of 105 patients (age: 67.8 ± 6.9 years; female/male, 39/66). A linear mixed-effects model using data for both eyes, with strict adjustments for sex, age, subjective refraction spherical equivalent, subjective refraction cylinder, corneal astigmatism, axial length, corneal higher-order aberrations and pupil diameter, ensured statistical validity. Uncorrected near visual acuity, corrected near visual acuity, and near spectacle independence were significantly better in the ZMB00 group (p<0.00068, Wald test) than in the ZXR00V group. Contrast sensitivity (visual angle of the test target: 4.0°/2.5°/1.6°/1.0°/0.7°) and contrast sensitivity with glare (4.0°/2.5°/1.6°/1.0°/0.7°) were significantly better in the ZXR00V group (p<0.00068, Wald test) than in the ZMB00 group. Uncorrected intermediate visual acuity, contrast sensitivity with glare (6.3°), and 25-item National Eye Institute Visual Function Questionnaire (VFQ-25) scores for General Vision were slightly but significantly better in the ZXR00V group than in the ZMB00 group (p<0.05, Wald test). At high-performance levels, the two IOL groups had different characteristics regarding various visual performance parameters.

## Introduction

Cataract surgeries are increasingly common worldwide, particularly in developed countries with aging populations [1, 2]. Surgical advancements, such as intraocular lenses with enhanced materials, have improved postoperative satisfaction and reduced the invasiveness of the

**Funding:** The authors received no specific funding for this work.

**Competing interests:** The authors have declared that no competing interests exist.

procedures [3–7]. Diffractive bifocal lenses were initially developed over 15 years ago, with the one-piece acrylic version providing improved near visual acuity and a reduced need for glasses [8]. However, these lenses have some negative effects, which motivated the development of extended-depth-of-focus lenses with no loss of intermediate vision or significant change in contrast sensitivity [9–11].

In Japan, where near vision is highly valued, +4 D diffractive bifocal lenses have been the mainstay of the market [12]. Their use has continued despite some issues; whether extended-depth-of-focus lenses can overcome these problems remains an important clinical question.

This study compares the clinical results of +4 D diffractive bifocal lenses (TECNIS® Multifocal IOL +4.0 D, ZMB, Johnson & Johnson Surgical Vision, Inc., Santa Ana, CA, USA) and extended-depth-of-focus lenses (TECNIS® Symfony, ZXR00V, Johnson & Johnson Surgical Vision, Inc., Santa Ana, CA, USA) to provide valuable insight for clinicians and enable them to make informed decisions regarding intraocular lens selection in cataract surgery. By analyzing a large sample of patients, the study seeks to overcome the limitations of previous research, which often included small patient populations and did not implant the same lens type in both eyes [13–15].

This study aims to identify each lens type's potential advantages and disadvantages through the evaluation of visual acuity, quality of vision, and patient satisfaction.

## Materials and methods

### Design

Retrospective comparative case series.

### Setting

Ophthalmology, Tsukazaki Hospital, Japan.

### Patients

We reviewed a consecutive case series of cataract patients who underwent bilateral implantation of Tecnis bifocal IOLs (ZMB00) and Tecnis EDOF IOLs (ZXR00V) from August 11, 2011, to February 22, 2020, with the right and left lenses implanted within 3 months of each other in the same way as in our previous study [9, 16]. Participants were recruited for enrollment in this consecutive case series study (outpatients with or without doctor referral). There was no potential self-selection bias likely to impact the results. The exclusion criteria were a history of other ocular diseases that could affect visual function, |subjective equivalent (SE)| > 2.00 D, |subjective refraction cylinder (CYL)| > 3.00 D and |corneal astigmatism (keratometric cylinder)| > 3.00 D at 10 weeks after surgery.

### Preoperative examination

Preoperative examinations were performed in the same way as in our previous study [9, 16]. All patients received full ophthalmologic examinations, including evaluations of the corneal curvature radius, corneal astigmatism, axial length, refractive status, ocular aberrations, pupil diameter, distance/intermediate/near visual acuity, contrast sensitivity, and contrast sensitivity under glare, as well as anterior segment evaluations using a slit lamp, tonometry, and indirect fundoscopy. The quality of vision was evaluated using the Japanese version of the 25-item National Eye Institute Visual Function Questionnaire (VFQ-25) [17]. The VFQ-25 was administered by experienced technicians or nurses in a face-to-face setting. Spectacle use was also

evaluated by inquiring how often the patient used spectacles for distance, intermediate and near vision (with possible responses of 'never,' 'sometimes' or 'always').

Uncorrected distance visual acuity (UDVA) and corrected distance visual acuity (CDVA) were measured at 5.0 m. Uncorrected intermediate visual acuity (UIVA) and corrected intermediate visual acuity (CIVA) were measured at 0.5 m. Uncorrected near visual acuity (UNVA) and corrected near visual acuity (CNVA) were measured at 0.3 m. Visual acuity was measured using the decimal visual acuity chart, and the measured decimal values were converted to the logarithm of the minimum angle of resolution (logMAR) scale. The corneal curvature radius, corneal astigmatism, and objective refractive status were measured using a KR-8900 autorefractor keratometer (Topcon, Tokyo, Japan). Axial length was measured using IOL Master 700 (Carl Zeiss Meditec AG, Jena, Germany) and AL-3000 (TOMEY, Nagoya, Japan) biometers. Contrast sensitivity and contrast sensitivity under glare (visual angle of the test target: 6.3°/4.0°/2.5°/1.6°/1.0°/0.7°; 13 contrast levels: 0.01 to 0.64 contrast or 2.00 to 0.34 $\log_{10}CS$) were measured using a CGT-1000 contrast glare tester (Takagi Seiko, Nakano, Japan) [9, 10, 16], and the pupil diameter and ocular aberrations were measured using a KR-1W Wavefront Analyzer (Topcon, Tokyo, Japan). Experienced technicians obtained all measurements.

## IOLs and surgical technique

Two types of diffractive lenses from Johnson & Johnson Vision, Tecnis Symfony EDOF IOL (ZXR00V) and Tecnis Multifocal +4.0 D (ZMB00) were examined in the same way as in our previous study [9, 16]. Both have acrylic optics measuring 6.0 mm in diameter (more precisely, the optic sizes of these lenses increase or decrease with IOL power and are mostly less than 6.0 mm), a biconvex optical structure (the lenses use a lenticular edge to reduce center thickness), and an aspheric optical section designed to suppress corneal spherical aberration. ZMB00 is a clear lens, and ZXR00V is a yellow lens with the same coloring as ZCB00V (OptiBlue®, Johnson & Johnson Surgical Vision, Inc. New Brunswick, NJ, USA), which has been reported to have a minor effect on visible wavelengths [18]. The ZXR00V IOL contains an achromatic diffraction surface to correct for corneal chromatic aberration. The ZMB00 IOL is a double-focus diffractive lens with +4 D near power [19–22]. The patients chose to undergo implantation with either bifocal or EDOF IOLs after they had been informed of the advantages and disadvantages associated with each type. The patients in the bifocal group received Tecnis bifocal IOLs (ZMB00) bilaterally, while those in the EDOF group received Tecnis EDOF IOLs (ZXR00V) bilaterally. The goal for all eyes was emmetropia for distant vision. Cataract surgeries were performed by 15 experienced cataract surgeons using the same standard technique of sutureless microincision phacoemulsification and the same protocol. The surgical procedures consisted of topical anesthesia, the creation of a scleral or corneal incision of 1.8 to 2.8 mm, 5 mm of continuous capsulorhexis, phacoemulsification cataract extraction and IOL implantation with an injector.

## Postoperative examination

The patients were evaluated at 10 weeks postoperatively. The postoperative examination protocol at 10 weeks was identical to the preoperative protocol.

## Statistical analyses

The sample size was calculated for an alpha of 0.00068 and a power of 0.80. A standard deviation in the VA of 0.10 logMAR units was presumed in addition to a minimum detectable difference of 1 line of VA (0.1 logMAR), based on our previous study [9]. This calculation

recommended the inclusion of 39 eyes per group. The bifocal group comprised 1326 eyes of 663 patients, and the EDOF group comprised 210 eyes of 105 patients; thus, the sample size was sufficient.

Similar to our previous study [9, 16], the two groups (bifocal IOL and EDOF IOL) were compared in terms of the following postoperative parameters at 10 weeks after surgery on both eyes: (1) mixed-effects linear regression: visual acuity (uncorrected/corrected, distance/intermediate/near), contrast sensitivity (with/without glare), and higher-order aberrations (ocular/internal, scaled to a pupil size of 4 mm/6 mm); (2) linear regression model or logistic regression: VFQ-25 scores; and (3) cumulative logistic regression: spectacle dependence (distance/intermediate/near). Analyses in both groups were adjusted for age, sex, axial length, subjective refraction spherical equivalent, subjective refraction cylinder, corneal astigmatism, corneal higher-order aberrations and pupil diameter. In regression analyses (2) and (3), the data were divided into two parts (left-eye data and right-eye data), and the regression model was applied to each dataset. Since discrete scores were observed for "peripheral vision", "color vision", "driving daytime", "driving nighttime", and "driving adverse conditions" on the VFQ-25, we treated them as binary data. We divided the patients into two groups (those with scores of 75 or lower and those with scores above 75) and applied the logistic regression model to these groups. The threshold was determined from the distribution of the following variables: Peripheral_Vision, Color_Vision, Driving_Daytime, Driving_Nighttime, and Driving_Adverse_Conditions. A threshold of 75 was used in this study because most VFQ-25 scores are >75 after surgery. The results of the left- and right-eye analyses were combined using the inverse variance method; the corrected values were calculated for the left- and right-eye datasets, and the average values were used. In the regression analysis, the Wald test was applied to evaluate the significance of differences in postoperative parameters between the two groups, and the significance level was set to 0.00068 after a Bonferroni correction. Correlation analysis between postoperative parameters was applied for the bifocal and EDOF groups, and a heatmap of Pearson's correlation coefficients was generated for each group. In the correlation analysis, two-sided t tests were used to evaluate whether the correlation coefficients were significantly different from zero, and the significance level was set to 0.00002 after a Bonferroni correction.

The statistical analyses were performed by using a commercially available software program (R, version 3.6.1; R Core Team, 2019, Vienna, Austria) [23].

### Ethics statement

This study conformed to the tenets of the Declaration of Helsinki and was approved by the Ethics Committee of Tsukazaki Hospital. All research was performed in accordance with relevant guidelines/regulations. Written informed consent was obtained from each subject. This study was registered as UMIN000035630: "Performance comparison among different intraocular lenses in cataract surgery."

## Results

### Patient characteristics

The patient demographics and pre/postoperative visual parameters are shown in S1A Table. The study enrolled 1536 eyes of 768 patients. The bifocal (ZMB00) group comprised 1326 eyes of 663 patients (67.0 ± 7.8 years; female/male, 518 [78.1%]/145 [21.9%]), and the EDOF (ZXR00V) group comprised 210 eyes of 105 patients (67.8 ± 6.9 years; female/male, 39 [37.1%]/66 [62.9%]).

## Comparison of postoperative parameters between Tecnis bifocal IOLs (ZMB00) and Tecnis EDOF IOLs (ZXR00V)

Multiple regression analysis was applied to all postoperative parameters of the bifocal (ZMB00) and EDOF (ZXR00V) groups at 10 weeks after surgery in both eyes in the same way as in our previous study [9, 16]; the parameters were adjusted by multiple regression with the explanatory variables in S1B Table, and the results of the analysis are shown in S2 Table. Uncorrected near visual acuity (UNVA), corrected near visual acuity (CNVA), and near spectacle independence were significantly better in the ZMB00 group than in the ZXR00V group ($p < 0.00068$, Wald test) (Table 1 and Fig 1). Contrast sensitivity (visual angle of the test target: 4.0°/2.5°/1.6°/1.0°/0.7°) and contrast sensitivity with glare (4.0°/2.5°/1.6°/1.0°/0.7°) were significantly better in the ZXR00V group than in the ZMB00 group ($p < 0.00068$, Wald test), and uncorrected intermediate visual acuity (UIVA), contrast sensitivity with glare (6.3°), and the National Eye Institute VFQ-25 scores for general vision were slightly better in the ZXR00V group than in the ZMB00 group ($p < 0.05$, Wald test) (Table 1 and Figs 1–3).

**Table 1. Parameters that demonstrated a significant difference at $p < 0.00068$ or $p < 0.05$ between the bifocal and EDOF groups at 10 weeks after surgery on both eyes.**

| Response, post | After adjustment | | Coefficient (95% CI) | p value (Wald test) |
|---|---|---|---|---|
| | ZMB00 | ZXR00V | | |
| UIVA | 0.20 ± 0.13 | 0.15 ± 0.10 | -0.06 (-0.11, 0.01) | 2.380E-02* |
| UNVA | 0.09 ± 0.10 | 0.38 ± 0.17 | 0.31 (0.26,0.35) | 6.138E-32** |
| CNVA | 0.02 ± 0.06 | 0.18 ± 0.20 | 0.15 (0.11,0.19) | 1.923E-12** |
| Contrast sensitivity | | | | |
| C_4.0 | 0.04 ± 0.01 | 0.03 ± 0.01 | -0.01 (-0.02, 0.01) | 4.004E-05** |
| C_2.5 | 0.06 ± 0.02 | 0.04 ± 0.02 | -0.02 (-0.03, 0.01) | 7.833E-06** |
| C_1.6 | 0.10 ± 0.03 | 0.07 ± 0.04 | -0.04 (-0.05, 0.02) | 3.243E-06** |
| C_1.0 | 0.20 ± 0.07 | 0.15 ± 0.09 | -0.07 (-0.10, 0.04) | 1.103E-05** |
| C_0.7 | 0.39 ± 0.09 | 0.30 ± 0.11 | -0.10 (-0.14, 0.06) | 4.372E-07** |
| Contrast sensitivity with glare | | | | |
| G_6.3 | 0.04 ± 0.03 | 0.02 ± 0.01 | -0.01 (-0.02, 0.00) | 1.388E-02* |
| G_4.0 | 0.06 ± 0.03 | 0.03 ± 0.02 | -0.02 (-0.03, 0.01) | 1.016E-05** |
| G_2.5 | 0.08 ± 0.04 | 0.04 ± 0.02 | -0.03 (-0.05, 0.02) | 9.385E-06** |
| G_1.6 | 0.14 ± 0.07 | 0.08 ± 0.05 | -0.05 (-0.08, 0.03) | 7.639E-06** |
| G_1.0 | 0.27 ± 0.10 | 0.15 ± 0.08 | -0.12 (-0.16, 0.08) | 2.012E-10** |
| G_0.7 | 0.42 ± 0.08 | 0.33 ± 0.10 | -0.10 (-0.13, 0.07) | 2.029E-08** |
| VFQ-25 | | | | |
| General vision | 80.37 ± 3.15 | 82.93 ± 2.66 | 4.12 (0.85,7.39) | 1.345E-02* |
| Spectacle dependence | | | | |
| Near | 83/7/1 | 19/7/21 | 3.27 (2.40,4.13) | 1.060E-13** |

Each parameter was adjusted by multiple regression with the explanatory variables in S1B Table. For each response variable, the mean and standard deviation for each numerical parameter or the counts for each categorical parameter (spectacle dependence: never/sometimes/always), the regression coefficient, its 95% confidence interval (CI), and the p value (Wald test) are shown.

C: contrast sensitivity; CNVA: corrected distance visual acuity; G: contrast sensitivity under glare; UIVA: uncorrected intermediate visual acuity; UNVA: uncorrected near visual acuity.

\* $p < 0.05$, \*\* $p < 0.00068$

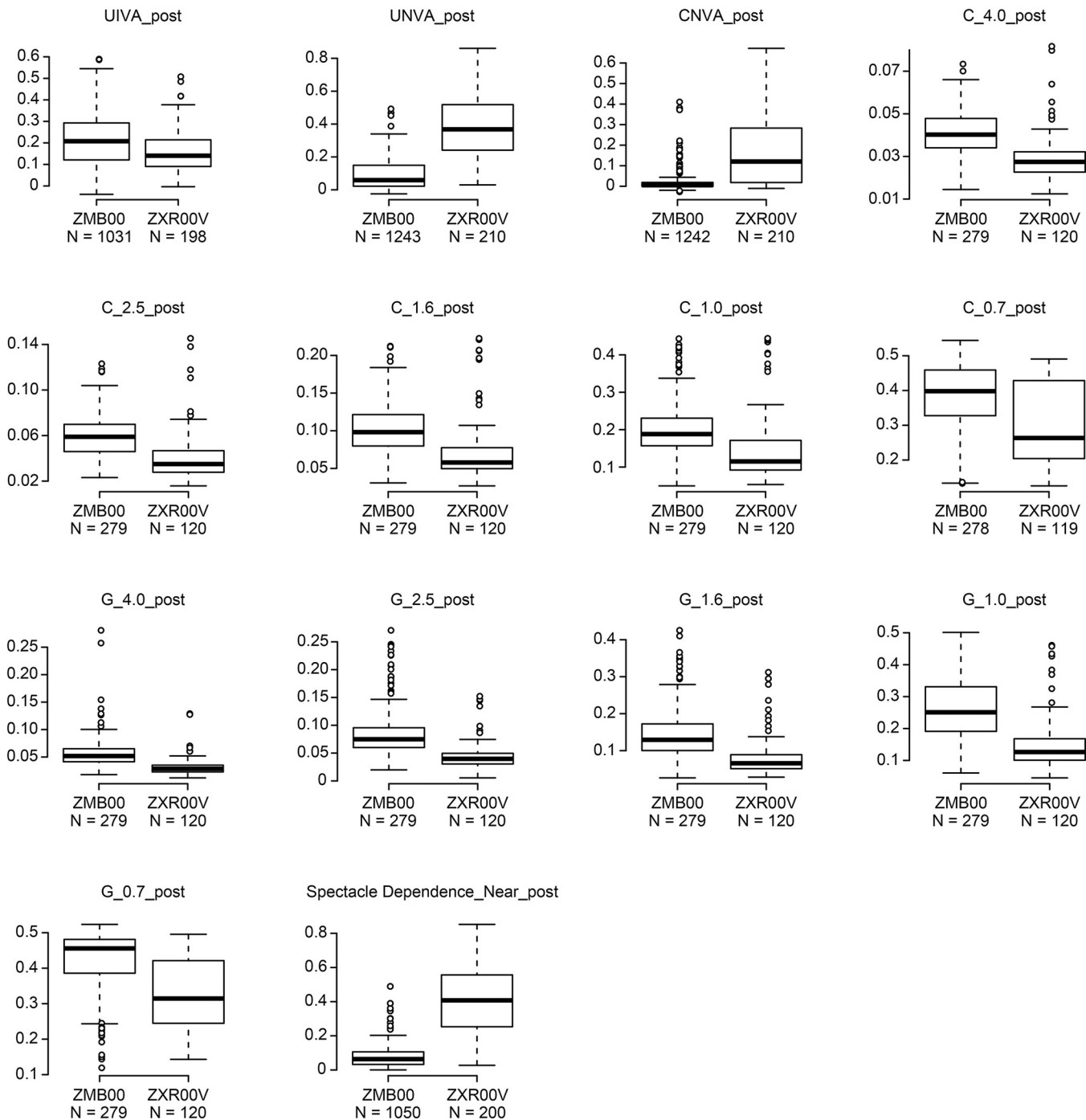

**Fig 1. Significantly different parameters between the bifocal and EDOF groups 10 weeks after bilateral eye surgery.** The line inside the box represents the median. To highlight suspected outliers, the upper whisker is set as the maximum or the third quartile+ 1.5 × IQR. The lower whisker indicates the minimum or the first quartile-1.5 × IQR. Each parameter was adjusted by multiple regression with the explanatory variables in S1B Table. The two-sided Wald test was performed to evaluate the significance of differences between the two groups, and the significance level was set to 0.00068 using a Bonferroni correction. An asterisk * indicates a significant difference between the two groups with p<0.00068. UIVA: uncorrected intermediate visual acuity; UNVA: uncorrected near visual acuity; CNVA: corrected near visual acuity; C: contrast sensitivity; G: contrast sensitivity with glare.

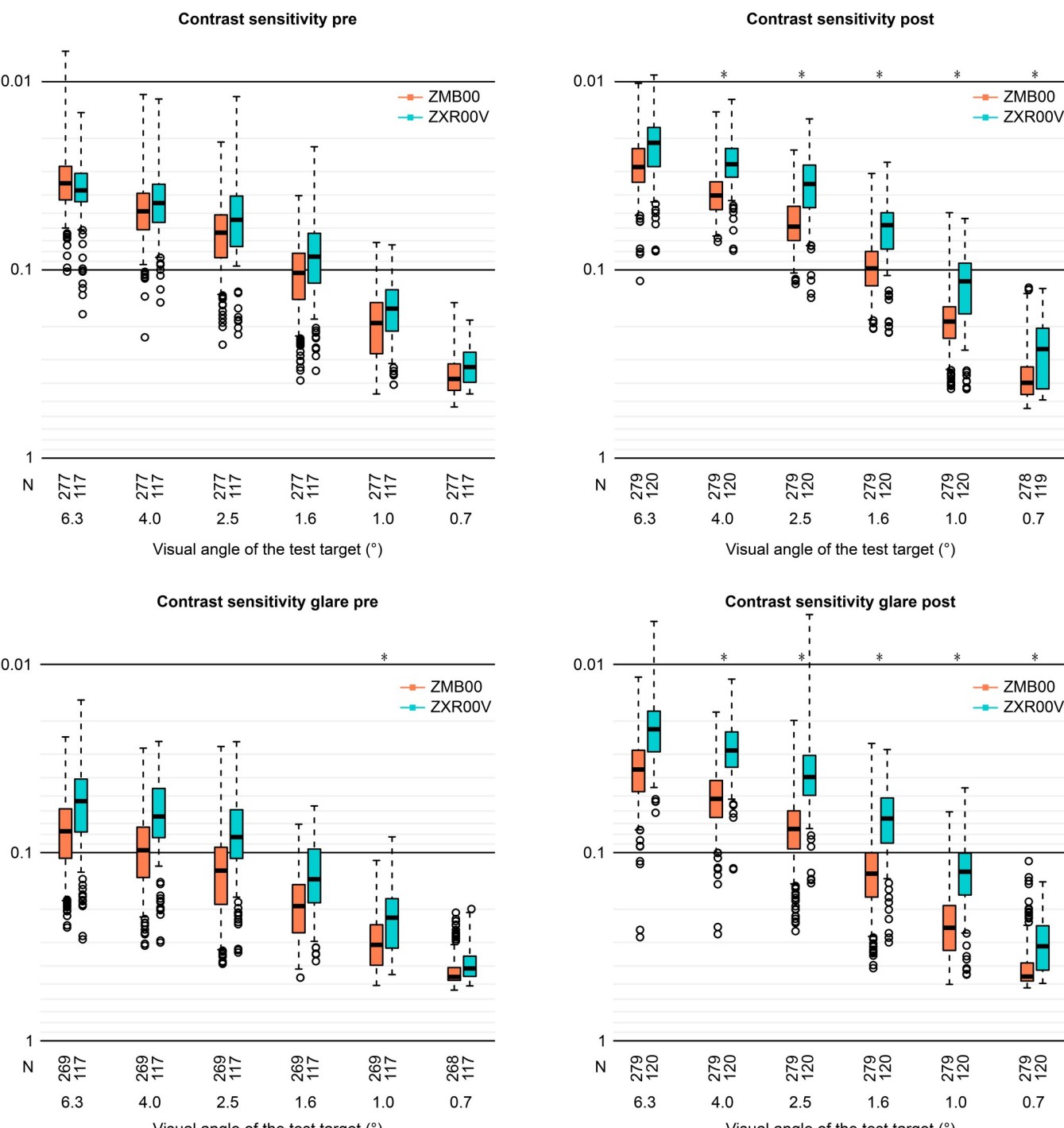

**Fig 2. Contrast sensitivity with/without glare in the bifocal/EDOF groups before/10 weeks after bilateral eye surgery.** In the box-and-whisker plots, the bottom of the box indicates the first quartile, and the top of the box indicates the third quartile. The band inside the box represents the median. To highlight suspected outliers, the upper whisker is set as the maximum or the third quartile+ 1.5 × IQR. The lower whisker indicates the minimum or the first quartile- 1.5 × IQR. Each parameter was adjusted by multiple linear regression with the explanatory variables in S1B Table. The two-sided Wald test was conducted to evaluate the significance of differences between the two groups, and the significance level was set to 0.0083 after a Bonferroni correction.

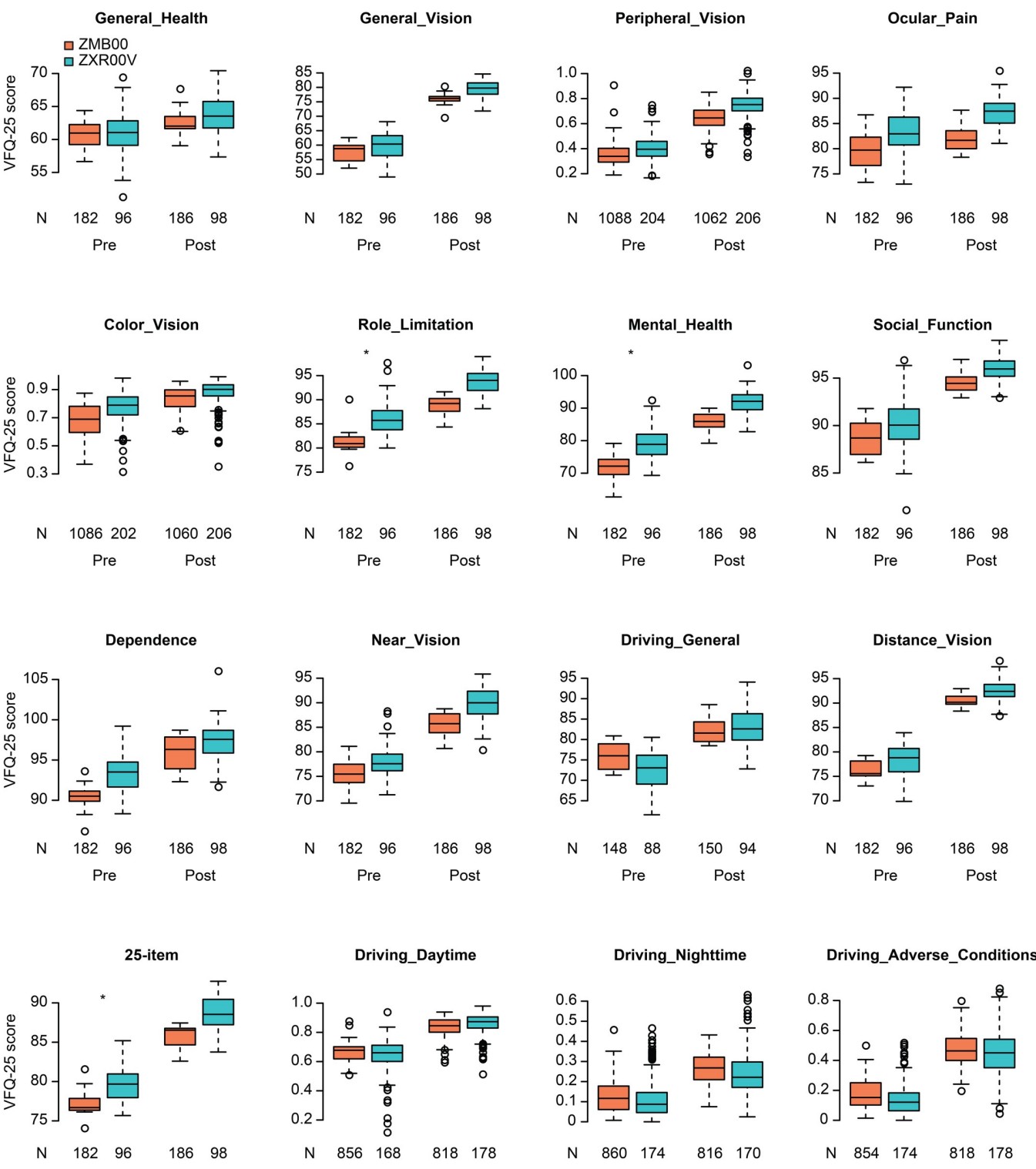

**Fig 3. VFQ-25 scores in the bifocal/EDOF groups before/10 weeks after bilateral eye surgery.** In the box-and-whisker plots, the bottom of the box indicates the first quartile, and the top of the box indicates the third quartile. The band inside the box represents the median. To highlight suspected outliers, the upper whisker is set as the maximum or the third quartile+ $1.5 \times IQR$. The lower whisker indicates the minimum or the first quartile-$1.5 \times IQR$. Each parameter was adjusted by multiple linear regression with the explanatory variables in S1B Table. The two-sided Wald test was conducted to evaluate the significance of differences between the two groups, and the significance level was set to 0.003125 after a Bonferroni correction. The asterisk * indicates a significant difference between the two groups with p<0.003125. In this figure, the predicted values for the following variables were the probabilities of whether the score was over 75

points: Peripheral_Vision, Color_Vision, Driving_Daytime, Driving_Nighttime, and Driving_Adverse_Conditions. Therefore, the y-axis scale for these variables ranges from 0 to 1.

## Discussion

In this study, we examined a large sample of patients with the same lens inserted in each eye using a wide range of items, including questionnaires related to visual function and eyeglass usage rates, in addition to visual function test values and anatomical measurements. We also performed a comprehensive comparative analysis using a linear mixed-effects model of data for both eyes, with strict adjustments for sex, age, subjective refraction spherical equivalent, subjective refraction cylinder, corneal astigmatism, axial length, corneal higher-order aberrations and pupil diameter.

We found that ZMB has a better uncorrected near visual acuity than Symfony and, accordingly, a lower dependence on glasses. The study by Liu et al. [14] (which directly compares ZMB and Symfony) concluded that there was no difference in spectacle dependency. The different conclusion based on the results in our study (lower dependence on glasses for near vision with ZMB) might be due to a larger sample size and the identification of a significant difference. Good near vision and low dependence on glasses with ZMB are consistent with previous reports [24–29]. There have been some reports that Symfony provides good near vision and a high rate of spectacle independence [30, 31], but many studies comparing Symfony with trifocal lenses have shown that the former provides significantly inferior near vision [32–35]. In fact, ZMB and Symfony are considered bifocal lenses based on an optical bench picture from the University of Heidelberg; the ZMB lens has +4.0 D near addition, while the Symfony has +1.75 D near addition, equaling a near focus of 34 cm and 78 cm, respectively, in an average-sized eye. Thus, Symfony has a clear disadvantage when tested at short distances. On the other hand, it was also clear that Symfony has fewer contrast-related problems than ZMB. Symfony's high contrast sensitivity has been noted in optical studies, and the same is true for good intermediate vision [36]. Regarding the contrast sensitivity of ZMB, some reports have shown that it is in the normal range in bright environments [10, 37], but it was inferior to that of Symfony in this study. Both uncorrected near visual acuity and contrast sensitivity are important factors that affect the way we see, and the clarification of the differences in clinical characteristics between ZMB and Symfony is a result that provides a basic decision-making tool in daily clinical practice when selecting a lens that meets each patient's needs.

There was no difference in higher-order aberrations (HOAs) between the two lenses; both lenses have a complex conic surface that is meant to reduce spherical aberration based on Liou & Brennan's eye model. This feature does not "suppress" other HOAs; in fact, it can induce asymmetric aberrations if not well centered and has a modulation transfer function (MTF) inferior to those of more moderately shaped IOLs [38, 39]. Although wavefront measurements on multifocal IOLs have been performed in many studies, one should approach these measurements with caution, keeping in mind how the multiple wavefronts of each IOL may affect the results.

One point clarified in this study that was not noted in previous reports is that there was no difference between the two lenses regarding the scores on the driving items. The fact that the VFQ-25 driving items were not significantly different between ZMB and Symfony, which has better contrast sensitivity than ZMB, reveals a potential problem in selecting Symfony. ZMB is associated with a significantly lower postoperative satisfaction rate than monofocal lens with regard to night driving [9]. It was previously demonstrated that Symfony insertion patients have a high complaint rate when driving at night [33]. As shown in S2 Fig, Symfony shows a

significant negative correlation between higher internal HOAs (excluding spherical aberrations) and the VFQ score for nighttime driving. This correlation is observed at a pupil diameter of 6 mm, which is considered the usual diameter of the pupil at night. Regarding this connection, it has been noted that Symfony has slightly larger aberrations than other EDOF lenses with the new design at large pupil diameters [40, 41]. Based on these results, Symfony could be cautiously indicated, especially for patients who drive at night.

It is presumed that Symfony's significantly better results in general vision, one of the items in VFQ-25, were contributed by its significantly higher contrast sensitivity and better intermediate vision. This is possibly because both contrast sensitivity and intermediate vision are essential for the comfortable viewing of computers, televisions, and other visual information devices that are indispensable in modern life. Although there was no difference in the overall mean scores on the VFQ-25, the results regarding good general vision may be important information for lens selection if the patient is not very particular about living a near-vision–free life and does not drive frequently at night. On the other hand, the mean scores on the VF-14 [42] reported by Liu et al. [14] were higher with Symfony than with ZMB. The correlation between the mean VFQ-25 and VF-14 scores has been reported to be approximately 0.6 [43], and it is statistically plausible that our results may differ from those of Liu et al. [14].

Several comparative studies between Symfony and other EDOF lenses and trifocal lenses have been reported [34, 44–51]. What they all have in common is that there is no one lens that outperforms the others in all the various visual parameters (visual acuity at various distances from far to near, contrast sensitivity, reading index, questionnaire, aberrations, etc.). Therefore, techniques such as placing lenses with different functions in each eye or shifting the postoperative target refractive values in each eye have been tried [52–54]. However, the sample size in each report has been possibly insufficient, and there are differences among the patient's characteristics; thus, there is currently no concrete consensus on a method.

There are some potential limitations of this study. One is that intermediate visual acuity and near visual acuity were measured only at 50 cm and near visual acuity at 30 cm, respectively, in the same way described in our previous study [9, 16]. Although there has been perpetual controversy about the examination distance [55], ideally, we should measure visual acuities at various distances to evaluate lens performance. However, in the present study, as the practical distance at examination has been reported to be essential, we adopted the distances that are within arm's reach when the patient holds a test chart. As Japanese people have a relatively short average height and arm length compared with European and American people, intermediate visual acuity was measured at 50 cm, which was assumed to be within the reach of most patients in the same way described in our previous study [9, 16]. On the other hand, Symfony is constructed for intermediate distances typical of desktop computer use (70–80 cm) and might not work very well at 50 cm. This should be considered when interpreting the results of our study.

Second, potential differences in the patients' backgrounds, including socioeconomic status, might exist between the two groups in this study. To increase the validity of the study, this large-scale, single-center study was performed under a consistent protocol as described in our previous study [9, 16], i.e., we evaluated the same series of pre- and postoperative comprehensive visual parameters, including the VFQ-25 score, after written informed consent was obtained from all the patients before surgery. Although this study was retrospective, each patient who underwent lens implantation was randomly and independently sampled, and all endpoints were measured. Furthermore, we strictly adjusted for age, sex, axial length, subjective refraction SE, subjective refraction CYL, corneal astigmatism (keratometric cylinder), corneal higher-order aberrations and pupil diameter in the same way described in our previous study [9, 16]. Because the data contain a mixture of items evaluated either in both eyes or in

each eye separately, we performed an analysis using a linear mixed model that accounted for bias and corrected for multiple observations for each eye per patient. It is common in statistics to assume that random assignment does not bias the results of the analysis, even if there are differences in the number of patients, such as the 1:n allocation used in clinical trials.

Third, this study was conducted at a single Japanese facility, and the VFQ-25 results might be different among different races, i.e., Japanese patients might tend to value near vision [12]. However, in other words, the results of a study involving patients from around the world may not be optimal for a particular region. In some studies involving individuals of the same race, the diversity of individuals could be a background factor suitable for adjustment when evaluating pure lens performance differences.

Fourth, the scientific rationale is weaker than that of a randomized prospective study. However, a randomized clinical trial for the selection of IOLs would be unethical and unrealistic, as the risk of replacement is high. Statistical studies such as this study, which are based on a large sample of cases and detailed clinical data, have been shown to have a certain degree of credibility, and we believe that they have value for clinical use.

## Conclusions

In conclusion, we compared the visual performance of a bifocal intraocular lens (ZMB00) and an EDOF IOL (ZXR00V). Uncorrected near visual acuity, corrected near visual acuity, and near spectacle independence were significantly better in the bifocal group, while contrast sensitivity (4.0˚/2.5˚/1.6˚/1.0˚/0.7˚) and contrast sensitivity with glare (4.0˚/2.5˚/1.6˚/1.0˚/0.7˚) were significantly better in the EDOF group. At high performance levels, the two IOL groups had different characteristics regarding various visual parameters. As a large-scale clinical research facility, we will continue to collect as much comprehensive data as possible in daily clinical practice and continue to use statistical methods to conduct one-on-one clinical comparative lens performance studies that can be used as a reference for lens selection.

## Supporting information

**S1 Table.** Patient demographics and pre/postoperative visual parameters (A, B). For categorical data, each category and its count and frequency are shown, and the two-sided Fisher's exact test was used to compare categorical data across the bifocal and EDOF IOL groups. For numerical data, the means and standard deviations are shown, and the two-sided Mann–Whitney test was used to compare numerical data across the bifocal and EDOF IOL groups. Parameters in the bifocal and EDOF groups used to adjust the linear regression model are shown in (B). Age, sex, axial length (at the time of surgery), subjective refraction spherical equivalent (SE), subjective refraction cylinder (CYL), corneal astigmatism (keratometric cylinder), corneal higher-order aberrations (astigmatism, total higher-order aberration (HOA), third-, fourth-, trefoil, coma, tetrafoil, second-order astigmatism (2nd Astig), and spherical, scaled to a pupil size of 4 mm/6 mm). CYL, subjective refraction cylinder; HOA, higher-order aberration; WF_4_post_C, wavefront_4_post_corneal; SE, subjective refraction spherical equivalent. (ZIP)

**S2 Table. Results of multiple regression analyses for all postoperative parameters of the bifocal (ZMB00) and EDOF (ZXR00V) groups 10 weeks after surgery in both eyes.** For numerical parameters, multiple mixed linear regression or multiple linear regression was applied, and cumulative logistic regression was applied to the spectacle dependence parameters. In the multiple linear regression or cumulative logistic regression, the variables in S1B Table were used as the explanatory variables. For each response variable, the regression

coefficient, its 95% confidence interval, and the p value (Wald test) are shown in (A). The original and corrected values (i.e., before and after adjusting with multiple linear regression) of the mean and standard deviation for each numerical parameter and the counts for each categorical parameter (spectacle dependence: never/sometimes/always), regression coefficient, 95% confidence interval, and p value (Wald test) are shown in (B).
(XLSX)

**S3 Table.** Pearson's correlation coefficients (A) and p values from the correlation analysis conducted with the two-sided t test (B) for all possible combinations of postoperative parameters, which were adjusted by multiple regression with the explanatory variables in S1B Table, in the bifocal (ZMB00) group. The sample size for calculating the correlation coefficients is shown in (C).
(XLSX)

**S4 Table.** Pearson's correlation coefficients (A) and p values from the correlation analysis conducted with the two-sided t test (B) for all possible combinations of postoperative parameters, which were adjusted by multiple regression with the explanatory variables in S1B Table, in the EDOF (ZXR00V) group. The sample size for calculating the correlation coefficients is shown in (C).
(XLSX)

**S1 Fig. Heatmap of Pearson's correlation coefficients between all possible combinations of variables in the ZMB00 group.** Pearson's correlation coefficients were adjusted by multiple regression with the explanatory variables in S1B Table. The asterisk * in this figure indicates a significant correlation between two parameters at $p < 0.00002$ after a Bonferroni correction. A two-sided t test was conducted to evaluate the significance of differences between the two groups. The sample size for each parameter is shown in S3C Table. The illustration was created using a commercially available software program (https://cran.r-project.org/web/packages/pheatmap/pheatmap.pdf) [23].
(TIF)

**S2 Fig. Heatmap of Pearson's correlation coefficients between all possible combinations of variables in the ZXR00V group.** Pearson's correlation coefficients were adjusted by multiple regression with the explanatory variables in S1B Table. The asterisk * in this figure indicates a significant correlation between two parameters at $p < 0.00002$ after a Bonferroni correction. The two-sided t test was applied to evaluate the significance of differences between the two groups. The sample size for each parameter is shown in S3C Table. The illustration was created using a commercially available software program (https://cran.r-project.org/web/packages/pheatmap/pheatmap.pdf) [23].
(TIF)

## Acknowledgments

We thank all the staff of Tsukazaki Hospital who were involved in this study. The statistical results and the validity of the study design were rigorously reviewed by a professional data analysis company in Japan (StaGen Co., Ltd.), and a certificate of the review of statistical methods by a professional statistician was issued. Research Square has also thoroughly confirmed the quality of this manuscript and has issued a validity report in the form of a Methods and Data Reporting Certification. The manuscript was edited for proper English language, grammar, punctuation, spelling, and overall style by one or more of the highly qualified native English-speaking editors at American Journal Experts (AJE), and a certificate was issued.

## Author Contributions

**Conceptualization:** Hitoshi Tabuchi, Hirotaka Tanabe, Kosuke Takase, Tomohiro Shojo, Tomofusa Yamauchi.

**Data curation:** Hirotaka Tanabe, Kosuke Takase, Tomohiro Shojo.

**Formal analysis:** Hirotaka Tanabe.

**Investigation:** Hirotaka Tanabe, Tomoki Shirakami, Kosuke Takase, Tomohiro Shojo.

**Methodology:** Hirotaka Tanabe.

**Project administration:** Hitoshi Tabuchi, Hirotaka Tanabe.

**Resources:** Hitoshi Tabuchi, Hirotaka Tanabe.

**Software:** Hitoshi Tabuchi, Hirotaka Tanabe.

**Supervision:** Hitoshi Tabuchi, Hirotaka Tanabe.

**Validation:** Hirotaka Tanabe, Kosuke Takase, Tomohiro Shojo, Tomofusa Yamauchi.

**Visualization:** Hirotaka Tanabe.

**Writing – original draft:** Hitoshi Tabuchi, Hirotaka Tanabe.

**Writing – review & editing:** Hirotaka Tanabe.

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
