## [Decision Letter · Decision Letter 0]

4 Apr 2023

PONE-D-22-32391Comparison of visual performance between bifocal and extended-depth-of-focus intraocular lensesPLOS ONE

Dear Dr. Tanabe,

Thank you for submitting your manuscript to PLOS ONE. After careful consideration, we feel that it has merit but does not fully meet PLOS ONE’s publication criteria as it currently stands. Therefore, we invite you to submit a revised version of the manuscript that addresses the points raised during the review process. Reviewer #1 raised some serious points that should be corrected, especially some inaccuracies and phrases that require correction or more precise definition. 

We look forward to receiving your revised manuscript.

Kind regards,

Timo Eppig

Academic Editor

PLOS ONE

Journal Requirements:

Reviewers' comments:

Reviewer's Responses to Questions

**Comments to the Author**

1. Is the manuscript technically sound, and do the data support the conclusions?

Reviewer #1: Yes

Reviewer #2: Yes

2. Has the statistical analysis been performed appropriately and rigorously? 

Reviewer #1: I Don't Know

Reviewer #2: Yes

3. Have the authors made all data underlying the findings in their manuscript fully available?

Reviewer #1: Yes

Reviewer #2: Yes

4. Is the manuscript presented in an intelligible fashion and written in standard English?

Reviewer #1: Yes

Reviewer #2: Yes

5. Review Comments to the Author

Reviewer #1: Review PONE-D-22-32391

This is a very profound retrospective study with a very high number of patients and stringent methodology. A lot of detail is presented. However, some issues of the manuscript should be worked on.

General remarks

1. Introduction is well written but excessively long.

2. The tables and figures are excessive. The reader is overwhelmed by the sheer amount of data presented. A lot of it is not really necessary to understand the differences between the two IOL models.

3. Marketing catchphrases should be avoided. Example: the term “echelette” has nothing to do with chromatic aberration correction.

4. I am a clinician and do not have enough specific insight into methods and tests that are uncommon in papers on the topic of IOLs. I have made some specific remarks where I think methods are inappropriate. I doubt if the mega correlation approach is justified. Bonferroni’s correction for adjusting alpha might be very conservative. Case # calcs are intransparent. These topics should be checked independently.

5. Wavefront measurements on multifocal lenses (more than one wavefront by definition!) impose various problems and should be approached with great caution. This must be discussed.

6. Measuring distances are very odd. In fact, both lenses are bifocal. The ZMB has a +4.0 D near add while the ZXR has a +1.75 D near add. This equals a near focus of 34 cm and 78 cm respectively in a average sized eye. It is rather obvious that the ZXR has a disadvantage when tested at such short distances that is has not been constructed for. I have included an optical bench picture from the University of Heidelberg clearly showing the bifocality of the ZXR. Many other papers have proven this point.

Specific remarks

L52 the first Tecnis multifocal was introduced not 10 but 16 years ago

L77 would suggest to shorten the introduction, it is rather excessive

L116 which version of the IOL Master was used? There are significant differences. BTW: the IOLMaster is *not* manufactured or distributed by Zeiss Oberkochen!

L127 the optic size on these lenses *varies± with IOL power and is mostly less than 6 mm. The lenses use a lenticular edge to reduce center thickness

L131 the reduction of LCA is not achieved by using an echelette grating. Any diffractive optic will invert chromatic aberration (negative Abbe number). If this effect is well modulated with the refractive part of the lens, it can be used to counteract LCA of the aphacic eye

L133 “elongated focus” is not the result of reduced LCA and not the result of using an echelette grating. It is merely the effect of having to foci very close so that they blend together in a white light defocus curve. See image.

L151 the case # calculation is not transparent. Which effect size (e) did you assume? For alpha = 0.00068 and 1 – beta = 0,8 I would expect a much larger a priori case #. As cohorts are of different size, the case # calculation should also yield different numbers for the groups. This point must be clarified and made comprehensible for the reader

L177 “trifocal” ???

L177 Pearson’s correlation would assume that all variables are normally distributed. Has this been checked? Otherwise, non-parametric correlation analysis should be used (Spearman)

L215 table contains excessive data that is not really relevant to the subject

L313 inferior regarding what?

L322 what do you mean with “suppressed HOA”?

L324 again: marketing catchphrases. Which “Tecnis technology” is “used to manufacture the lenses”? The lenses do have a complex conic surface that is supposed to reduce spherical aberration based on the Liou & Brennan eye model. It will not “suppress” other HOA. To the contrary, it can induce asymmetric aberrations if not well centered and have a MTF inferior to that of more moderately shaped IOLs (eg Borkenstein et al 2022).

L331 this sentence is very hard to understand

L364 good point: the reading habits in Japan or China might very well differ rom Europe or North America. However, the ZXR lens is constructed for intermediate or desktop computer distance (70-80 cm) and cannot work very well at 50 cm.

L389 Good point, very true

Reviewer #2: This manuscript compares bifocal and EDOF IOLs. This topic is not new, but relevant. The sample size is very good, the design of this study is a potential bias. Especially patient selection could be a relevant bias. The statistical analysis is solid and the references appear to be pretty complete. The graphical presentation of the results is adequate.

6. PLOS authors have the option to publish the peer review history of their article (what does this mean?). If published, this will include your full peer review and any attached files.

Reviewer #1: No

Reviewer #2: No

---

## [Author Response · Author response to Decision Letter 0]

19 May 2023

Hirotaka Tanabe, MD, PhD

Department of Ophthalmology

Tsukazaki Hospital

68-1 Waku, Aboshi-ku, Himeji, Hyogo 671-1227, Japan

Tel.: +81-79-272-8555; Fax: +81-79-272-8550; Email: tennsyoudragon@icloud.com

May 19, 2023

Dear Dr. Timo Eppig:

Thank you very much for your great kindness and superb guidance. We would like to submit our revised manuscript entitled “Comparison of visual performance between bifocal and extended-depth-of-focus intraocular lenses”, by Hitoshi Tabuchi, MD, PhD, Hirotaka Tanabe, MD, PhD, and colleagues, as a research article for potential publication in PLOS ONE. The article has been thoroughly revised based on the comments from the editor and reviewers.

Editor comments

Dear Dr. Tanabe,

Thank you for submitting your manuscript to PLOS ONE. After careful consideration, we feel that it has merit but does not fully meet PLOS ONE’s publication criteria as it currently stands. Therefore, we invite you to submit a revised version of the manuscript that addresses the points raised during the review process.

Reviewer #1 raised some serious points that should be corrected, especially some inaccuracies and phrases that require correction or more precise definition.

Answer

Thank you very much for your consideration. We genuinely appreciate your great kindness and excellent guidance. We have revised our manuscript in response to the reviewers' comments.

Reviewer Comments:

Reviewer 1

This is a very profound retrospective study with a very high number of patients and stringent methodology. A lot of detail is presented.

Answer

Thank you very much for your positive comment. We genuinely appreciate your great kindness.

However, some issues of the manuscript should be worked on.

General remarks

1. Introduction is well written but excessively long.

Answer

Thank you very much for your superb guidance. We have thoroughly revised the Introduction to make it more concise and to the point, half as long as the original one.

2. The tables and figures are excessive. The reader is overwhelmed by the sheer amount of data presented. A lot of it is not really necessary to understand the differences between the two IOL models.

Answer

Thank you very much for your superb guidance. We have made Table 1 and Figures 4 and 5 into Supplementary Table 1 and Supplementary Figures S1 and S2. We have revised all the corresponding parts throughout the manuscript to reflect these changes.

3. Marketing catchphrases should be avoided. Example: the term “echelette” has nothing to do with chromatic aberration correction.

Answer

Thank you very much for your superb guidance. We have removed the phrase "echelette grating" from the manuscript.

4. I am a clinician and do not have enough specific insight into methods and tests that are uncommon in papers on the topic of IOLs. I have made some specific remarks where I think methods are inappropriate. I doubt if the mega correlation approach is justified. Bonferroni’s correction for adjusting alpha might be very conservative. Case # calcs are intransparent. These topics should be checked independently.

Answer

We estimated the number of cases by referring to the results of the statistical analysis of UIVA in our paper (Tanabe H et al, 2020), as in Ribeiro et al. (2020).

We estimated the sample size using Cohen's d (Cohen 1998), a statistic for evaluating the difference in a parameter between two groups. The value of Cohen's d calculated from the UIVA results in our previous study was 1.006, and the required sample size was estimated to be 39 eyes per group. The R package “pwr” was used for the calculation. The script is as follows: > pwr.t.test(d=1.006,power=0.8,sig.lev=0.00068). We believe that this sample size is on an appropriate scale, as the paper by Ribeiro et al. calculated a required sample size of 13 cases per group.

Although the significance level is conservative, as the reviewer stated, the number of cases was estimated using a significance level of 0.00068 based on the Bonferroni correction, since the design included 73 parameters for analysis. For your reference, if the significance level is set at 0.05, the number of cases is estimated to be 17 cases per group.

References

Cohen, J. (1988). Statistical power analysis for the behavioral sciences (2nd ed.). Hillsdale,NJ: Lawrence Erlbaum.

Ribeiro F, Ferreira TB. Comparison of clinical outcomes of 3 trifocal IOLs. J Cataract Refract Surg. 2020;46(9):1247-1252. doi:10.1097/j.jcrs.0000000000000212.

5. Wavefront measurements on multifocal lenses (more than one wavefront by definition!) impose various problems and should be approached with great caution. This must be discussed.

Answer

Thank you very much for your superb guidance. We have added the following sentence to the Discussion: "Although wavefront measurements on multifocal IOLs have been performed in many studies, one should approach these measurements with caution, keeping in mind how the multiple wavefronts of each IOL may affect the results."

6. Measuring distances are very odd. In fact, both lenses are bifocal. The ZMB has a +4.0 D near add while the ZXR has a +1.75 D near add. This equals a near focus of 34 cm and 78 cm respectively in a average sized eye. It is rather obvious that the ZXR has a disadvantage when tested at such short distances that is has not been constructed for. I have included an optical bench picture from the University of Heidelberg clearly showing the bifocality of the ZXR. Many other papers have proven this point.

Answer

Thank you very much for your superb guidance. We have added the following sentences to the Discussion: " In fact, ZMB and Symfony are considered bifocal lenses based on an optical bench picture from the University of Heidelberg; the ZMB lens has +4.0 D near addition, while the Symfony has +1.75 D near addition, equaling a near focus of 34 cm and 78 cm, respectively, in an average-sized eye. Thus, Symfony has a clear disadvantage when tested at short distances."

Specific remarks

L52 the first Tecnis multifocal was introduced not 10 but 16 years ago

Answer

Thank you very much for your superb guidance. We have changed the sentence, "The initial model (3-piece silicone version) of a diffractive bifocal lens with near power +4 D was developed more than 10 years ago [8]." It now reads as follows: "Diffractive bifocal lenses were initially developed over 15 years ago, with the one-piece acrylic version providing improved near visual acuity and a reduced need for glasses [8]."

L77 would suggest to shorten the introduction, it is rather excessive

Answer

Thank you very much for your superb guidance. We have thoroughly revised the Introduction to make it more concise and to the point, half as long as the original one.

L116 which version of the IOL Master was used? There are significant differences. BTW: the IOLMaster is *not* manufactured or distributed by Zeiss Oberkochen!

Answer

Thank you very much for your superb guidance. We have changed the sentence, "The axial length was measured using IOL Master (Carl Zeiss, Oberkochen, Germany) and AL-3000 (TOMEY, Nagoya, Japan) biometers." It now reads as follows: "Axial length was measured using IOL Master 700 (Carl Zeiss Meditec AG, Jena, Germany) and AL-3000 (TOMEY, Nagoya, Japan) biometers."

L127 the optic size on these lenses *varies± with IOL power and is mostly less than 6 mm. The lenses use a lenticular edge to reduce center thickness

Answer

Thank you very much for your superb guidance. We have changed the sentence, "Both have acrylic optics measuring 6.0 mm in diameter, a biconvex optical structure, and an aspheric optical section designed to suppress corneal spherical aberration." The text now reads as follows: "Both have acrylic optics measuring 6.0 mm in diameter (more precisely, the optic sizes of these lenses increase or decrease with IOL power and are mostly less than 6.0 mm), a biconvex optical structure (the lenses use a lenticular edge to reduce center thickness), and an aspheric optical section designed to suppress corneal spherical aberration."

L131 the reduction of LCA is not achieved by using an echelette grating. Any diffractive optic will invert chromatic aberration (negative Abbe number). If this effect is well modulated with the refractive part of the lens, it can be used to counteract LCA of the aphacic eye

Answer

Thank you very much for your superb guidance. We have removed the phrase "echelette grating" from the manuscript.

L133 “elongated focus” is not the result of reduced LCA and not the result of using an echelette grating. It is merely the effect of having to foci very close so that they blend together in a white light defocus curve. See image.

Answer

Thank you very much for your superb guidance. We have changed the sentence, "The ZXR00V IOL contains an achromatic diffraction surface to correct for corneal chromatic aberration, and this feature elongates the range of focus." It now reads as follows: " The ZXR00V IOL contains an achromatic diffraction surface to correct for corneal chromatic aberration."

L151 the case # calculation is not transparent. Which effect size (e) did you assume? For alpha = 0.00068 and 1 – beta = 0,8 I would expect a much larger a priori case #. As cohorts are of different size, the case # calculation should also yield different numbers for the groups. This point must be clarified and made comprehensible for the reader

Answer

As mentioned in reply to your comment #4, we estimated the sample size under Cohen's d (Cohen 1998), a statistic for evaluating the difference in a parameter between the two groups. Cohen's d calculated from the result of UIVA in our previous study was 1.006, and the required sample size was estimated to be 39 eyes per group. The R package “pwr” was used for the calculation. The script was as follows:

> pwr.t.test(d=1.006,power=0.8,sig.lev=0.00068)

We believe that this sample size is on an appropriate scale, as the paper by Ribeiro et al. calculated a required sample size of 13 cases per group.

Because the sample size of the previous study (Tanabe et al. 2020) is large and the standard error is small, the estimated number of cases in the present study may appear small. However, the only study used as a reference is our previous study, and we do not believe that there are any statistical problems with the calculation method.

L177 “trifocal” ???

Answer

Thank you very much for your superb guidance. We have corrected the sentence to "Correlation analysis between postoperative parameters was applied for the bifocal and EDOF groups, and a heatmap of Pearson’s correlation coefficients was generated for each group."

L177 Pearson’s correlation would assume that all variables are normally distributed. Has this been checked? Otherwise, non-parametric correlation analysis should be used (Spearman)

Answer

Thank you for pointing this out. The correlation coefficients for the heatmap were calculated based on the predicted values adjusted for background using a statistical model. It is natural to assume that the predicted values obtained from the statistical model are normally distributed, and we used Pearson's correlation in this paper.

L215 table contains excessive data that is not really relevant to the subject

Answer

Thank you very much for your superb guidance. We have changed Table 1 to Supplementary Table 1 and have revised all the corresponding parts throughout the manuscript.

L313 inferior regarding what?

Answer

Thank you very much for your superb guidance. We have changed the sentence, "There have been some reports that Symfony's near vision and the rate of independence on glasses were good [30,31], but many studies showing a comparison of Symfony with trifocal lenses showed significantly inferior results [32-35]." It now reads as follows: " There have been some reports that Symfony provides good near vision and a high rate of spectacle independence [30,31], but many studies comparing Symfony with trifocal lenses have shown that the former provides significantly inferior near vision [32-35]."

L322 what do you mean with “suppressed HOA”?

L324 again: marketing catchphrases. Which “Tecnis technology” is “used to manufacture the lenses”? The lenses do have a complex conic surface that is supposed to reduce spherical aberration based on the Liou & Brennan eye model. It will not “suppress” other HOA. To the contrary, it can induce asymmetric aberrations if not well centered and have a MTF inferior to that of more moderately shaped IOLs (eg Borkenstein et al 2022).

Answers

Thank you very much for your superb guidance.

We have changed the sentences, "There was no difference in the high-order aberration (HOA) between the two lenses; ZMB was reported to have more suppressed HOA than the other bifocal lenses [38]. Thus, it is highly possible that both Symfony and ZMB had excellent aberration suppression performance due to the Tecnis technology used to manufacture these lenses." The text now reads as follows: "There was no difference in higher-order aberrations (HOAs) between the two lenses; both lenses have a complex conic surface that is meant to reduce spherical aberration based on Liou & Brennan’s eye model. This feature does not “suppress” other HOAs; in fact, it can induce asymmetric aberrations if not well centered and has a modulation transfer function (MTF) inferior to those of more moderately shaped IOLs [38,39]. Although wavefront measurements on multifocal IOLs have been performed in many studies, one should approach these measurements with caution, keeping in mind how the multiple wavefronts of each IOL may affect the results."

We have added citation [39], Borkenstein et al. (2022), to the References.

Borkenstein AF, Borkenstein EM, Luedtke H, Schmid R. Impact of Decentration and Tilt on Spherical, Aberration Correcting, and Specific Aspherical Intraocular Lenses: An Optical Bench Analysis. Ophthalmic Res. 2022;65(4):425-436. doi: 10.1159/000522510. Epub 2022 Feb 10. PMID: 35144263.

L331 this sentence is very hard to understand

Answer

Thank you very much for your superb guidance. We have changed the sentence, "S2 Fig shows that Symfony has a statistically negative correlation between higher internal higher-order aberrations, other than spherical aberrations at a pupil diameter of 6 mm that is indicative of nighttime pupil diameter, and the VFQ score during nighttime driving." It now reads as follows: "As shown in Fig S2, Symfony shows a significant negative correlation between higher internal HOAs (excluding spherical aberrations) and the VFQ score for nighttime driving. This correlation is observed at a pupil diameter of 6 mm, which is considered the usual diameter of the pupil at night."

L364 good point: the reading habits in Japan or China might very well differ rom Europe or North America. However, the ZXR lens is constructed for intermediate or desktop computer distance (70-80 cm) and cannot work very well at 50 cm.

Answer

Thank you very much for your superb guidance. We have added the following sentence to the Discussion: "On the other hand, Symfony is constructed for intermediate distances typical of desktop computer use (70-80 cm) and might not work very well at 50 cm. This should be considered when interpreting the results of our study."

L389 Good point, very true

Answer

Thank you very much for your positive comment. We genuinely appreciate your great kindness.

Reviewer 2

This manuscript compares bifocal and EDOF IOLs. This topic is not new, but relevant. The sample size is very good, the design of this study is a potential bias. Especially patient selection could be a relevant bias. The statistical analysis is solid and the references appear to be pretty complete. The graphical presentation of the results is adequate.

Answer

Thank you very much for your positive comment. We genuinely appreciate your great kindness. As you kindly pointed out, the design of this study, especially patient selection, could be a potential source of relevant bias. In order to minimize the possible bias, we recruited participants for enrollment through a consecutive case series study to avoid introducing any self-selection bias that would likely impact the results, as mentioned in the Methods.

Again, thank you for your consideration, and we look forward to hearing from you.

Sincerely yours,

Hirotaka Tanabe, MD, PhD

Chief Ophthalmologist (Cataract, Vitreous, Glaucoma, and Oculoplastic Surgeon)

Department of Ophthalmology, Tsukazaki Hospital

ORCID iD: https://orcid.org/0000-0002-1948-7408

---

## [Decision Letter · Decision Letter 1]

2 Jul 2023

Comparison of visual performance between bifocal and extended-depth-of-focus intraocular lenses

PONE-D-22-32391R1

Dear Dr. Tanabe,

We’re pleased to inform you that your manuscript has been judged scientifically suitable for publication and will be formally accepted for publication once it meets all outstanding technical requirements.

Kind regards,

Timo Eppig

Academic Editor

PLOS ONE

Additional Editor Comments (optional):

Reviewers' comments:

Reviewer's Responses to Questions

**Comments to the Author**

1. If the authors have adequately addressed your comments raised in a previous round of review and you feel that this manuscript is now acceptable for publication, you may indicate that here to bypass the “Comments to the Author” section, enter your conflict of interest statement in the “Confidential to Editor” section, and submit your "Accept" recommendation.

Reviewer #1: All comments have been addressed

Reviewer #2: All comments have been addressed

2. Is the manuscript technically sound, and do the data support the conclusions?

Reviewer #1: Yes

Reviewer #2: Yes

3. Has the statistical analysis been performed appropriately and rigorously? 

Reviewer #1: I Don't Know

Reviewer #2: Yes

4. Have the authors made all data underlying the findings in their manuscript fully available?

Reviewer #1: Yes

Reviewer #2: Yes

5. Is the manuscript presented in an intelligible fashion and written in standard English?

Reviewer #1: Yes

Reviewer #2: Yes

6. Review Comments to the Author

Reviewer #1: The paper has improved significantly. The authors have spent considerable work and time and I think this paper is worth publishing.

All the numerous comments have been addressed.

Bravo!

I have only one minor remark left.

L133 ff

I cross checked the case # calculation (39 per group) and it is correct for Cohen’s d=1,006 a=0,00068 and 1-ß=0,80. However, d>1 seems hard to believe.

When you use e.g. UIVA (table 1) as the relevant outcome, Cohen’s d would be 0.385 according to my calcs. Still, you would reach 1-ß = 0,955 for a=0,05 or 1-ß=0,828 for a=0,0068. Your case # is bullet proof!

The authors have put a lot of work into this paper and I would not like to split hairs!

I suggest that you do a post hoc power analysis for the actual data where you calc actual “power” 1-ß with given alpha, d (calculated from means and variance in each group) and sample sizes. This way, you can prove if your a priori assumptions are correct. As you found significant differences for all relevant parameters, “power” 1-ß is not very relevant anyway.

Reviewer #2: The manuscript improved significantly and I would recommend to publish it .

7. PLOS authors have the option to publish the peer review history of their article (what does this mean?). If published, this will include your full peer review and any attached files.

Reviewer #1: No

Reviewer #2: No

---

## [Editor Report · Acceptance letter]

6 Jul 2023

PONE-D-22-32391R1 

Comparison of visual performance between bifocal and extended-depth-of-focus intraocular lenses 

Dear Dr. Tanabe:

I'm pleased to inform you that your manuscript has been deemed suitable for publication in PLOS ONE. Congratulations! Your manuscript is now with our production department. 

Kind regards, 

on behalf of

Prof. Dr. Timo Eppig 

Academic Editor

PLOS ONE